# Application of Rosmarinic Acid with Its Derivatives in the Treatment of Microbial Pathogens

**DOI:** 10.3390/molecules28104243

**Published:** 2023-05-22

**Authors:** Ourdia-Nouara Kernou, Zahra Azzouz, Khodir Madani, Patricia Rijo

**Affiliations:** 1Laboratoire de Biomathématiques, Biophysique, Biochimie, et Scientométrie (L3BS), Faculté des Sciences de la Nature et de la Vie, Université de Bejaia, Bejaia 06000, Algeria; madani28dz2022@yahoo.fr; 2Laboratoire de Microbiologie Appliquée (LMA), Faculté des Sciences de la Nature et de la Vie, Université de Bejaia, Bejaia 06000, Algeria; zahraazzouz@yahoo.fr; 3Centre de Recherche en Technologie Agroalimentaire (CRTAA), Route de Targua-Ouzemour, Bejaia 06000, Algeria; 4CBIOS-Centro de Investigação em Biociências e Tecnologias da Saúde, Universida de Lusófona, Campo Grande 376, 1749-028 Lisbon, Portugal; p1609@ulusofona.pt; 5Instituto de Investigação do Medicamento (iMed.ULisboa), Faculdade de Farmácia, Universidade de Liboa, 1649-003 Lisboa, Portugal

**Keywords:** rosmarinic acid, antimicrobial resistance, synergistic effect, antibiofilm

## Abstract

The emergence of the antimicrobial resistance phenomena on and the harmful consequences of the use of antibiotics motivate the necessity of innovative antimicrobial therapies, while natural substances are considered a promising alternative. Rosmarin is an original plant compound listed among the hydroxycinnamic acids. This substance has been widely used to fight microbial pathology and chronic infections from microorganisms like bacteria, fungi and viruses. Also, various derivatives of rosmarinic acid, such as the propyl ester of rosmarinic acid, rosmarinic acid methyl ester or the hexyl ester of rosmarinic acid, have been synthesized chemically, which have been isolated as natural antimicrobial agents. Rosmarinic acid and its derivatives were combined with antibiotics to obtain a synergistic effect. This review reports on the antimicrobial effects of rosmarinic acid and its associated derivatives, both in their free form and in combination with other microbial pathogens, and mechanisms of action.

## 1. Introduction

Antimicrobial resistance (AMR) is a significant public health problem. According to ECDC and WHO estimates, more than 670,000 illnesses per year in the European area are caused by antibiotic-resistant bacteria, and over 33,000 human deaths are directly caused by these infections [1]. Antimicrobial resistance is directly related to the indiscriminate use of antibiotics and is developing at extremely high rates worldwide, including African countries like Cameroon [2]. Despite inadequate laboratory capacity to monitor AMR, the African continent shows evidence of the global trend of increasing drug resistance.

Some germs can not only spread in hospitals, but also in the great outdoors, where a significant resistance has been observed [3]. With increased drug consumption, increased hospital admissions, and increases in deaths, the financial consequences of AMR are financially devastating, including astronomical medical costs [4].

Antibiotic-resistant pathogenic bacteria include *P. aeruginosa*, *S. aureus*, *Enterobacteriaceae* and *Enterococcus* spp., while infections caused by nonresistant germs are significantly more difficult to treat [5].

In addition, the current overuse of conventional antibiotics and their improper use have reduced their efficacy, creating a serious problem for world health as well as development, and as such, the antimicrobial drug range is rapidly expanding compared to the available therapeutic treatments, making the latter ineffective or even inefficient [6]. The demand for new antimicrobial products is high, and products derived from natural sources are seen as a promising solution, including plant polyphenols, such as rosmarinic acid, which is a natural substance with antimicrobial activity.

## 2. Derivates of Rosmarinic Acid

Both 3,4-dihydroxyphenyllactic acid (DHPL) and caffeic acid (3,4- dihydroxycinnamic acid) are esterified to form rosmarinic acid (RA) (Figure 1). Scarpati and Oriente [7] were the first to determine the chemical structure of RA. They extracted rosemary RA from *Rosmarinus officinalis* from Lamiceae, and assigned the same name to rosmarinic acid. The tannin-like chemicals of the Lamiaceae have been called “Labiatengerbstoff” for a long time. However, not all species of Lamiaceae have RA, and “Labiatengerbstoff” may also have some other phenolic compounds. The plant family Boraginaceae also consistently contains RA. Other related caffeic acid esters as similar derivatives are in addition to RAs which are esters of caffeine and quinic acid including caffeoylshikimic and chlorogenic acids.

*Helicteresisora* has isolated isorinic acid (caffeoyl-4′-hydroxyphenyllactate), 4-O-glucosylated RA and 4/4′-O-diglucosylated RA [8], and also the cis-isomer of RA produced by *Salvia nemorosa* [9]. Many substances allegedly producing RA are reported, such as lithospermic acid from *Lycopus europaeus*, consisting of RA and caffeic acid [10] or *Lithospermum ruderale* [11], slithospermic acid B from *Salvia miltiorrhiza* (likewise called salvianolic acid B), which is composed of two RA molecules, and a number of other salvianolic acids [12,13], rabdosiin from *Rabdosia japonica* [14,15], sagecoumarin, melitric acid and sagerinic acid from *Salvia officinalis* [16,17], or yunnanic acids from *Salvia yunnanensis* [18] and several others reported in Bulgakov et al. [19]. RA methyl ethers and their derivatives are often reported, such as sage methylmelitric acid [16] and *Clerodendranthusspicatus* clerodendranoic acid [20].

The biosynthetic process of these more complex chemicals has not yet been studied, but their structures allow us to deduce that RA or derivatives of RA and other phenylpropanoids can be used for their production. There are more taxa with species that contain RA and related compounds in addition to the families Lamiaceae and Boraginaceae. The plants now recognized as having the “lowest” levels of RA are hornworts (Anthocerotaceae). In addition to RA, hornworts also contain lignan-like compounds that are associated with RA (such as anthocerotonic acid, megacerotonic acid, and anthocerodizonin) [21].

**Figure 1 molecules-28-04243-f001:**
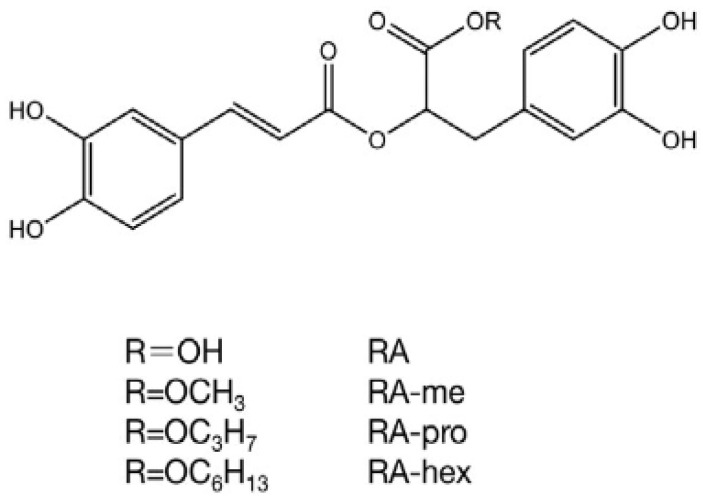
Variations in the structure of the acid rosmarinic and their derivatives. RA: rosmarinic acid, RA-hex: RA hexyl ester, RA-pro: RA propyl ester, RA-me: RA methyl ester [22].

The RA has been identified and isolated in 162 plants to date as a monomeric component, which are *Adenium obesum* [23], *Alkanna sfikasiana* Tan [24], *Anchusa azurea* [25], *Anchusa italica* [26], *Anchusa strigosa*, *Anthoceros punctatus* [27], *Apeiba tibourbou* [28], *Arctopus monacanthus* [29], *Arnebia purpurea* [30], *Baccharis chilco* [31], *Barbarea integrifolia* [32], *Bellis sylvestris* [33], *Blechnum Brasiliense* [34], *Canna edulis* [35], *Celastrus hindsii* [36], *Centella asiática* [37], *Chloranthus fortune* [38], *Chloranthus multistachys* [39], *Clerodendranthus spicatus* [40], *Clinopodium chinense* var. *Parviflorum* [41], *Clinopodium tomentosum* [42], *Clinopodium urticifolium* [43], *Coleus aromaticus* [44], *Coleus forskohlii* [45], *Coleus parvifolius* [46], *Colocasia esculenta* [47], *Cordia alliodora* [48], *Cordia bicolor* [49], *Cordia boissieri* [50], *Cordia dentata* [49], *Cordia latifolia* [51], *Cordia megalantha* [49], *Cordia morelosana* Standley [52], *Cordia sinensis* [53], *Cordia verbenácea* [54], *Cynoglossum columnae* [55], *Dracocephalum fruticulosum* [56], *Dracocephalum heterophyllum* [57], *Dracocephalum nutans* [56], *Dracocephalum palmatum* [58], *Dracocephalum tanguticum* [59], *Ehretia asperula* [60], *Ehretia obtusifolia* [61], *Ehretia philippinensis* [62], *Ehretia thyrsiflora* [63], *Elsholtiza bodinieri* [64], *Elsholtzia rugulosa* [65], *Elsholtzia splendens* [66], *Farfugium japonicum* [67], *Foeniculum vulgare* [68], *Forsythia koreana* [69], *Gastrocotyle hispida* [70], *Glechoma longituba* [71], *Hamelia patens* [72], *Hedera hélix* [73], *Helicteres angustifolia* [74], *Helicteres hirsuta* [75], *Helicteres isora* [8], *Hypenia salzmannii* [76], *Hyptis atrorubens* Poit. [77], *Hyptis capitata* [78], *Hyptis pectinata* [79], *Hyptis suaveolens* [80], *Hyptis verticillata* [81], *Hyssopus cuspidatus* [82], *Ipomoea turpethum* [83], *Isodon eriocalyx* [84], *Isodon flexicaulis* [85], *Isodon lophanthoides* var. *graciliflorus* [86], Isodon oresbius [87], *Isodon rubescens* [88], *Isodon rugosus* [89], *Isodon sculponeata* [90], *Keiskea japónica* [91], *Lallemantia iberica* [92], *Lavandula angustifolia* [93], *Lepechinia graveolens* [94], *Lepechinia meyenii* [95], *Lepechinia speciosa* [96], *Lycopus europaeus* [97], *Lycopus lucidus* [98], *Marrubium vulgare* [99], *Meehania urticifolia* [100], *Melissa officinalis* [101], *Mentha dumetorum* [102], *Mentha haplocalyx* [103], *Mentha longifolia* [104], *Mentha piperita* [105], *Mentha spicata* [106], *Mesona chinensis* [107], *Micromeria myrtifolia* [108], *Microsorum fortune* [109], *Momordica balsamina* [110], *Nepeta asterotricha* [111], *Nepeta cadmea* [112], *Nepeta curviflora* [113], *Ocimum campechianum* [114], *Ocimum sanctum* [115], *Origanum dictamnus* [116], *Origanum glandulosum* [117], *Origanum majorana* [118], *Origanum minutiflorum* [119], *Origanum rotundifolium* [120], *Origanum vulgare* [121], *Paris veriticillata* [122], *Perilla frutescens* [123,124], *Perilla frutescens* var. *acuta* [125], *Perovskia atriplicifolia* [126], *Plectranthus forsteri* [127], *Plectranthus hadiensis* var. *Tomentosus* [128], *Plectranthus madagascariensis* [129], *Plectranthus scutellarioides* [130], *Polygomun aviculane* [131], *Prunella vulgaris* [132], *Prunella vulgaris* var. *Lilacina* [133,134], *Quercus serrata* [135], *Rosmarinus officinalis* [136], *Salvia absconditiflora* [137], *Salvia castanea* [138], *Salvia cavaleriei* [139], *Salvia cerino-pruinosa* [140], *Salvia chinensis* [141,142], Salvia deserta Schang [143], *Salvia flava* Forrest [144], *Salvia grandifolia* [145], *Salvia kiaometiensis* Lévl. [146], *Salvia limbata* [147], *Salvia miltiorrhiza* [148], *Salvia officinalis* [16], *Salvia palaestina* [149], *Salvia plebeian* [150], *Salvia przewalskii* [151], *Salvia sonchifolia* [152], *Salvia splendens* Sellow [153], *Salvia trichoclada* [154], *Salvia viridis* [155], *Salvia trichoclada* [154], *Salvia viridis* [155], *Salvia yunaansis* [156], *Sanicula europaea* [157], *Sanicula lamelligera* [158], *Sarcandra glabra* [159], *Sideritis albiflora, Sideritis leptoclada* [160], *Solanum betaceum* [161], *Solenostemon monostachys* [162], *Symphytum officinale* [163], *Thunbergia laurifolia* [164], *Thymus alternans* [165], *Thymus atlanticus* [166], *Thymus praecox* sub *spgrossheimii* [167], *Thymus praecox* sub *spgrossheimii* [168], *Thymus quinquecostatus* var. japonica [169], *Thymus serpyllum* [170], *Thymus sibthorpii* Bentham [171], *Thymus sipyleus* subsp. Sipyleus var. sipyleus [172], *Thymus vulgaris* [173], *Tournefortia sarmentosa* [174], *Veronica sibirica* L. [175], *Ziziphora clinopodioides* [176], *Zostera marina* [177], and *Zostera noltii* [178].

## 3. Antimicrobial Activity

There have been hundreds of research studies conducted on the antimicrobial activity of AR (Table 1). In this article, we will mention the most recent studies conducted, specifically, RA is used as a natural phytogenic additive in animal and poultry nutrition to improve their overall health, performance measures, the digestive system’s structure and function and its potential to modify the intestinal microbiota and decrease the number of disease-causing bacteria such as *Salmonella* spp, *E. coli,* and several other species of harmful bacteria [179,180]. Rosemary extracts may contain RA as the primary bioactive antimicrobial agent. However, using the methanol extract containing about 30% carnosic acid, with 16% carnosol and 5% RA, Gram-positive and Gram-negative bacteria were shown to be sensitive to rosemary, making it an excellent antibacterial, in contrast to an aqueous extract containing 15% RA which had more limited effects [181].

Due to their intense antimicrobial properties, medicinal plants, herbs and their oils are attracting great interest as innovative and alternative drugs, such as RA [182].

According to Benedec et al. [183], RA showed better antioxidant activity in vitro (DPPH technique) as well as significant action towards the Gram-positive bacteria. In addition, *Rosmarinus officinalis* extract showed even greater inhibition of the growth of Gram-positive bacteria than the Gentamicin control (*Candida albicans*). On the other hand, these researchers noted that this extract was without effect toward Gram-negative bacteria such as *S. typhimurium*, *L. monocytogenes, E. coli*, *S. aureus,* and *C. albicans* were found to be resistant to RAs derived from: *Hyssopus officinalis* L., *M. officinalis* L., *O. vulgare* L. [183]. RA addition decreases the rate of mortality in Japanese encephalitis virus-infected mice. Compared to animals infected with no RA treatment, the viral load was greatly reduced (*p* < 0.001) in RA-treated infected rats 8–9 days after infection [184].

The antibacterial properties of tannic acid have long been recognized as effective against both methicillin-resistant *Staphylococcus aureus* and other microorganisms [185]. Currently, one of the molecules used as a target for antibacterial polymer applications is the tannic acid–polymer metal complex [186]. Hospitalized patients are severely harmed by *S. aureus*, and tannic acid is known to be an inhibitor of various resistance phenotypes of *S. aureus* [187]. Furthermore, by reducing cell counts and numbers, RA inhibits the development of *S. carnosus* LTH1502 and *E. coli* K-12 [188].

Moreno et al. [181] examined extracts of *Rosmarinus officinalis* through a combination of biological tests. Antimicrobial activities were analyzed by both disk diffusion and dilution broth techniques. Gram-positive bacteria, including *S. aureus*, *B. megaterium*, *B. subtilis*, and *E. faecalis*, were more sensitive to the methanolic extract, which contains 30% carnosic acid, 16% carnosol, and 5% rosmarinic acid (minimum inhibition concentration (MIC), 2 to 15 mg/mL). Gram-negative bacteria such as *K. pneumoniae, E. coli*, *X. campestris* pv. campestris, and *P. mirabilis* were also treated with MIC 2 to 60 mg/mL, as well as yeasts such as *S. cerevisiae*, *C. albicans*, and *P. pastoris* (MIC of 4 mg/mL). However, the aqueous extract with a 15% rosmarinic acid content only exhibited a narrow spectrum of activity. The MICs for methanolic and water extracts correlated significantly with the values for pure carnosic acid and rosmarinic acid. So, these results indicated a good performance in relation to the antimicrobial efficacy with rosemary extracts combined with the relevant phenolic extracts. The principal antimicrobial bioactive agents in rosemary extracts were suggested to be carnosic acid or rosmarinic acid. From the point of view of practicality, it could be considered as a good nutritional supplement and herbal pharmaceutical product.

Rosmarinic acid has antibacterial properties against *Staphylococcus aureus*, *E. coli*, *B. subtilis*, and *Salmonella*. Hayriye [189] tested the effect of natural phenolic compounds extracted from vegetables, fruits, herbs and spices against these pathogens and *E. coli* had minimum bactericidal concentrations (MBC) of 0.9 mg/mL and minimum inhibitory concentrations (MIC) of 0.8 mg/mL. *Salmonella* had MIC and MBC of 0.9 and 1.0 mg/mL, respectively. *Staphylococcus aureus* and *B. subtilis* had MIC and MBC values of 1.0 and 1.1 mg/mL [189], respectively.

The strains LM1, LM2, and LM3 of *L. monocytogenes* were analyzed, and the presence of rosmarinic acid was shown to have no antibacterial effect over the incubation period of 60 h [190]. Previously, rosmarinic acid has been shown to exhibit high susceptibility to Gram-negative bacteria when exposed to rosmarinic acid, after 60 h of incubation, Salmonella species showed substantial levels of antimicrobial resistance, and the MICs of rosmarinic acid for *S. enteridis*, *S. choleraesuis* subsp., and *S. paratyphi* were less than 20 ppm [190].

Furthermore, rosmarinic acid had previously been recognized as an anti-HIV drug capable of inhibiting HIV replication [191]. The discovery of nitro and dinitro-rosmarinic acids, which inhibit viral replication by blocking HIV-I integrase, has significantly enhanced the anti-HIV efficacy of rosmarinic acid [192].

**Table 1 molecules-28-04243-t001:** Rosmarinic acid and its derivatives are used as antibiotics against several pathogenic microorganisms.

Pathogenic Microorganisms	Active Concentrations	References	Pathogenic Microorganisms	Active Concentrations	References
*Staphylococcus epidermidis* 5001 *Stenotrophomonas maltophilia* *Enterococcus faecalis C159-6* *Staphylococcus lugdunensis T26A3* *Pseudomonas aeruginosa ATCC 27583*	MIC (0.3 mg/mL of RA) MIC (0.3 mg/mL of RA) MIC (0.3 mg/mL of RA) MIC (0.6 mg/mL of RA) MIC (2.5 mg/mL of RA)	[77]	*Escherichia coli*	MIC 0.8 mg/mL of RA; MBC 0.9 mg/mL of RA	[189]
			*Staphylococcus aureus*	MIC 1.0 mg/mL of RA; MBC 1.1 mg/mL of RA	
			*Salmonella*	MIC 0.9 mg/mL of RA; MBC 1.0 mg/mL of RA	
			*Bacillus subtilis*	MIC 1.0 mg/mL of RA; MBC 1.1 mg/mL of RA	
*Corynebacterium* T25-17 *Mycobacterium smegmatis* 5003 *Staphylococcus warneri* T12A12	MIC (2.5 mg/mL of RA) MIC (1.2 mg/mL of RA) MIC (1.2 mg/mL of RA)		*Micrococcus luteus*	MIC 0.1 mg/mL; MBC 0.2 mg/mL	[193]
			*Rothia mucilagenosa*	MIC 0.1 mg/mL; MBC 0.2 mg/mL	
*Klebsiella* sp.	IZ 28 mm at 1 mg/mL of RA	[177]	*Streptococcus agalactiae*	MIC 0.05 mg/mL; MBC 0.1 mg/mL	
*Stenotrophomonas maltophela*	IZ 19 mm at 1 mg/mL of RA		*Streptococcus angiosus*	MIC 0.05 mg/mL; MBC 0.1 mg/mL	
*Streptomyces* sp.	IZ 26 mm at 1 mg/mL of RA		*Streptococcus dysgalactie*	MIC 0.05 mg/mL; MBC 0.1 mg/mL	
*Pantoea agglomerans*	IZ 18 mm at 1 mg/mL of RA		*Streptococcus oralis*	MIC 0.05 mg/mL; MBC 0.1 mg/mL	
*Paenibacillus chibensis*	IZ < 1 mm at RA-methyl ester IZ 4.4 mm at tannic acid IZ > 2 mm at RA-hexyl ester IZ between 3 mm and 4 mm at RA-propyl ester	[194]	*Streptococcus parasanquinis*	MIC 0.05 mg/mL; MBC 0.1 mg/mL	
			*Streptococcus pyogenes*	MIC 0.1 mg/mL; MBC 0.2 mg/mL	
			*Streptococcus salivarius*	MIC 0.002 mg/mL; MBC 0.004 mg/mL	
*Staphylococcus waeneri*	IZ < 1 mm at RA-methyl ester IZ 5 mm at tannic acid IZ ˃ 2 mm at RA-hexyl ester IZ between 2 mm and 3 mm at RA-propyl ester		*Staphylococcus aureus*	MIC ˃ 0.8 mg/mL; MBC ˃ 0.8 mg/mL	
			*Staphylococcus hominis*	MIC 0.4 mg/mL; MBC 0.8 mg/mL	
*Bacillus cereus*	IZ > 3 mm at RA-methyl ester IZ 6 mm at tannic acid IZ 7.7 mm at RA-hexyl ester IZ 9 mm at RA-propyl ester		*Enterobacter cloacae*	MIC 0.1 mg/mL; MBC 0.2 mg/mL	
			*Strenotrophomonas maltophila*	MIC0.4 mg/mL; MBC 0.8 mg/mL	
*Bacillus subtilis*	MICs 5 ppm of AR	[181]	*Candida albicans* 475/15	MIC 0.1 mg/mL of RA MFC 0.2 mg/mL of RA	
*Bacillus cereus*	MICs 10 ppm of AR		*Candida albicans* 13/15	MIC 0.1 mg/mL of RA; MFC 0.2 mg/mL of RA	
*Bacillus polymyxa*	MICs 15 ppm of AR		*Candida albicans* 17/15	MIC 0.1 mg/mL of RA; MFC 0.2 mg/mL of RA	
*C. butyricum: C. sporogenes*	MICs of <20 ppm of RA	[190]	*Candida albicans* 527/14	MIC 0.15 mg/mL of RA; MFC 0.3 mg/mL of RA	
*SARS-CoV-2*	IC_50_ at 25.47 ng μL^−1^ of RA	[195]	*Candida albicans* 10/15	MIC 0.15 mg/mL of RA; MFC 0.3 mg/mL of RA	
*Enterovirus A71 (EV-A71)*	In vivo 100 mg/kg/day of RA	[196]	*Candida albicans* 532	MIC 0.1 mg/mL of RA; MFC 0.2 mg/mL of RA	
*S. aureus*	IZ 22 ± 1.00 mm at 1.33 ± 0.01 mg/g of RA	[183]	*Candida albicans* ATCC 10231	MIC 0.2 mg/mL of RA; MFC 0.4 mg/mL of RA	
*L. monocytogenes*	IZ 20 ± 2.00 mm at 1.33 ± 0.01 mg/g of RA		*Candia krusei* H1/16	MIC 0.2 mg/mL of RA; MFC 0.4 mg/mL of RA	
*E. coli*	IZ 8 ± 0.50 mm at 1.33 ± 0.01 mg/g of RA		*Candida glabrata* 4/6/15	MIC 0.1 mg/mL of RA; MFC 0.2 mg/mL of RA	
*S. typhimurium*	IZ 10 ± 0.00 mm at 1.33 ± 0.01 mg/g of RA		*Candida tropicalis* ATCC 750	MIC at 0.2 mg/mL of RA MFC at 0.4 mg/mL of RA	
*C. albicans*	IZ 28 ± 3.00 mm at 1.33 ± 0.01 mg/g of RA		*Candida parapsilosis* ATCC 22019	MIC at 0.1 mg/mL of RA MFC at 0.2 mg/mL of RA	

IZ: Inhibition Zone, MIC:minimal inhibitory concentration. MFC: minimal fungicidal concentration.

## 4. Antibiofilm Activity

The production of biofilms is one of the main processes responsible for antibiotic resistance. Recent research has revealed that natural substances based on secondary metabolites from plants can prevent the development of biofilms, which are responsible for about 80% of bacterial diseases [197,198]. Biofilms, which are bacterial colonies adhering to the surface and enveloped in a protective extracellular matrix, make bacteria up to 1000 times less susceptible to antibiotics and represent a real health problem [197]. The most common opportunistic fungal diseases in the world are Candida species, which form highly structured biofilms, which are collections of cells of different natures surrounded by an extracellular matrix. Furthermore, the current standard treatment for these infections is to seek innovative treatments for biofilm-related disorders, as these fungal biofilms are typically resistant to conventional antifungal drugs [198].

The quorum sensing inhibition (QSI) potential of rosmarinic acid (RA) towards *Aeromonas hydrophila* strains MTCC 1739, AH 1, and AH 12 was examined. The *A. hydrophila* pathogenic strains were isolated from infectious zebrafish species as well as an RA biofilm inhibitory concentration (BIC) versus *A. hydrophila* strains that was found as 750 μg mL^−1^. RA at this concentration decreased QS-induced production of hemolysin, elastase, and lipase from *A. hydrophila*. However, in FT-IR analysis, AR-treated *A. hydrophila* cells exhibited a reduction in cellular components, and the analysis of gene expression affirmed the negative regulation of virulence genes such as aerA, ahh1, ahyB, and lip. Zebrafish contaminated with *A. hydrophila* and given RA showed increased survival. Therefore, a study demonstrated the use of RA as an herbal compound to control biofilm formation by QS as well as virulence factor generation in *A. hydrophila* [199].

Biofilms of *C. krusei* H1/16 showed the highest resistance against rosmarinic acid treatment; MBEC > 1.6 mg/mL was the minimum biofilm eradication concentration, and biofilms of both *C. albicans* 475/15 and *C. albicans* ATCC 10,231 and eradicated with 0.4 mg/mL of rosmarinic acid. In contrast to cell attachment, biofilm formation was more strongly affected for *C. albicans* strains than for non-*C. albicans* [193].

RA consumption affected the formation of biofilms at a concentration- and the time-dependent manner, further implying for RA as an effective antimicrobial agent as well as for destroying the activity of planktonic cells and reducing the formation of biofilms at the earlier time stage to their development [200]. RA also inhibits the growth of *E. coli* K-12 and *S. carnosus* LTH1502, reducing the density and number of cells [188]. In acidic medium, RA was found to react chemically to nitrite ions to generate 6,6-nitro and 6-dinitrorosmarinic acids, the latter were active at submolecular levels as HIV-1 integrase inhibitors and inhibited viral replication in MT-4 cells, and antiviral effects [192]. RA nitration significantly increased integrase inhibition and antiviral effects without increasing the levels of cellular toxicity. In addition, RA also possesses antimicrobial effects against lactic acid bacteria, yeasts, molds, *Enterobacteriaceae* spp, and *Pseudomonas* spp, as well as against psychotropic drugs and *L. monocytogenes* from chicken meat [201]. In addition, RA exhibits inhibitory effects on the *S. aureus* cocktail by intimating morphological changes, decreasing and reducing all viable cells, and inducing morphological alterations into cheese and meat samples, from cell shrinkage to the formation of burr-like structures on the cell surface [202,203,204].

The antibacterial effects of rosmarinic acid (RA) against clinical strains of *S. aureus* from catheter infections were tested by Slobodníková et al. [200]. The regeneration method detected 24 h biofilm eradication activity on microtiter plates. The microtiter plate approach permitted the quantification for biofilm formation activity following application of RA to bacterial samples at 0, 1, 3, and 6 h post biofilm formation, with RA exhibiting antimicrobial activity at concentrations ranging from 625 to 1250 g·mL^−1^ (MICs equal to MBCs). In the concentration of the 156 to 5000 g·mL^−1^ evaluated range, there were no biofilm eradicating actions on the 24 h biofilm. When processed at the beginning of biofilm formation, RA subinhibitory doses inhibited the synthesis of biofilm; in concentrations less than the subinhibitory level, the formation of biofilm mass was increased in a time- and concentration-dependent manner. This evidence indicates the potential for RA to be an effective topical antimicrobial agent for treating catheter-related infections, with activity against both planktonic forms of bacteria and inhibitory activity during the early stages of biofilm development. However, it is not practical to use RA as the only agent to treat catheter-related infections [205].

## 5. Modes of Action

RA action modes are numerous (Figure 2), including immunomodulatory, analgesic, antimicrobial, neuroprotective, anticancer, anti-inflammatory, antioxidant, and anti-Alzheimer effects, and fertility stimulators [179,206,207,208,209,210,211,212,213,214,215].

The mode of action of RA may involve inactivation of cellular enzymes and changes in membrane permeability [216]. Zhu et al. [194] examined the reaction of RA-hex to glucosidase using the MB docking technique as a model study to explain the mechanism related to the inhibitory activity of *S. cerevisiae* glucosidase. In general, the authors found that the dihydroxyphenyl and hexyl groups are required for interaction with the active site (Figure 3).

In addition, RA allows depolymerization of the cell membrane [217], and proteomics of the bacterial membrane will reveal a modification after contact with RA [77]. RA has a bacteriostatic effect, capable of destroying bacterial cells and their proteins as well as blocking Na+/K+-ATPase activity in the cell [189]. In addition, rosmarinic acid specifically inhibits the P-protein of hepatitis B virus [218], RA could influence the early state of viral infection and directly affect viral particles by affecting virus-P-selectin glycoprotein ligand-1 (PSGL1) interactions with heparan sulfate substance without affecting virus–scavenger receptor B2 (SCARB2) interactions [196].

Among the antifungal mechanisms of rosmarinic acid, there was a decrease in mitochondrial activity, deterioration of membrane integrity, and mild inhibition of virus activity, integrity, and a slight suppression in proteases production except ergosterol binding. Antibiofilmic activity was associated to a limited extent with a decrease in the production of exopolysaccharides [193].

**Figure 2 molecules-28-04243-f002:**
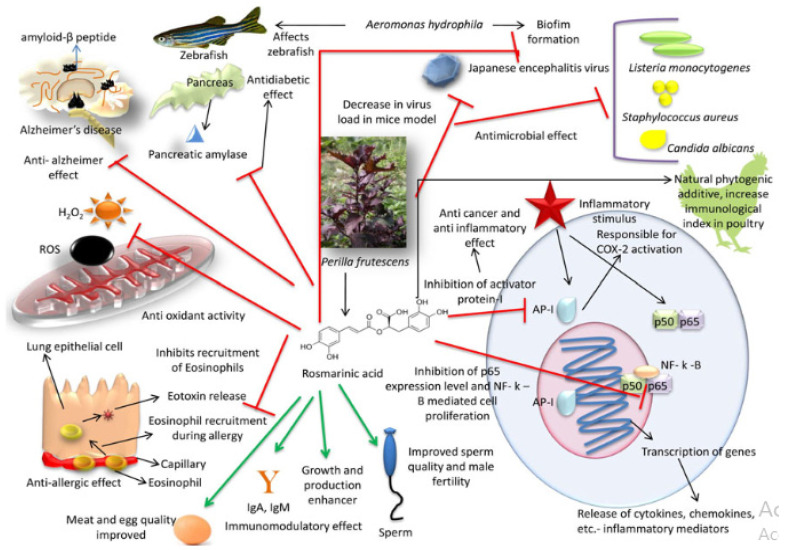
Mode of action with a positive impact of rosmarinic acid [219].

**Figure 3 molecules-28-04243-f003:**
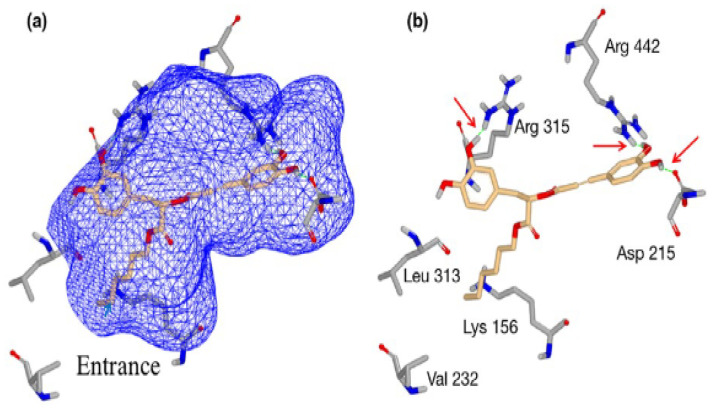
A molecular docking system using rosmarinic acid hexyl ester (RA-hex) to the α-glucosidase active site from *Saccharomyces cerevisiae*, (**a**) RA-hex incorporated into the enzymatic pocket; (**b**) RA-hex interactions with key amino acid residues [194].

## 6. Combined Application of Rosmarinic Acid and Derivatives with Other Antimicrobial Agents

The main issue facing contemporary medicine: microbial resistance is being addressed from several angles, one of these approaches involves the use of new drugs in synergistic combination with conventional antibiotics that are already administered in treatment and to which microbial resistance has already evolved (Table 2). With penicillin, *Polyalthia longifolia* extracts have demonstrated a synergistic antibacterial action against a clinical isolate of MRSA, blocking the formation of biofilms, and the way that a substance works is connected to a phenomenon of cell agglutination and an alteration of the integrity of the cell membrane, which leads to cell lysis. These discoveries all indicate that extracts of *P. longifolia* leaves are a valuable source of trustworthy chemicals for the creation of new antimicrobials to combat antibiotic resistance [220].

*S. aureus* strains were discovered to be sensitive to RA’s antibacterial properties, with minimum inhibitory concentrations for methicillin-resistant *S. aureus* (MRSA) and *S. aureus* found to be 10 mg/mL and 0.8, respectively. Furthermore, RA showed synergistic benefits against *S. aureus* with amoxicillin, ofloxacin, and vancomycin drugs, but only with vancomycin against MRSA. In treatment with RA in conjunction with antibiotics, effectiveness is more than the use of individual antibiotics, according to a time-kill study. In addition, when RA and vancomycin are used together, the expression of the adhesion protein MSCRAMM (Microbial Surface Components Recognizing Adhesive Matrix Molecules) in MRSA and *S. aureus* is significantly reduced compared to RA alone [182].

The study’s findings by Coiai et al. [221] indicate that the antimicrobial powers of RA and ulvane, as well as their environmental impact, could be used in many products, especially since the COVID-19 pandemic made this need widespread. This would meet both the health and environmental needs of environmentalists.

Under in vitro conditions, RA has synergistic efficacy with antibiotics such as vancomycin to destroy methicillin-resistant *S. aureus* (MRSA) [182].

In this regard, it has been recommended to further exploit the antimicrobial characteristics of RA to discover useful applications for sustainable development. According to the research of Lu et al. [222], RA binds to VmFbpA, the FbpA of *V. metschnikovii*, both more strongly and competitively than Fe^3+^, with a KD value of 8 M vs. 17 M. Moreover, at a concentration of 1000 M, RA was able to reduce the growth of *V. metschnikovii* by up to 1/3 compared to controls. It was interesting to note that sodium citrate (SC), although not in turn a growth inhibitor, enhanced the impact of RA on growth. In addition to complete inhibition of *V. metschnikovii* growth at 100/100 M, the RA/SC mixture completely inhibits the growth of *V. vulnificus* and *V. parahaemolyticus*, at concentrations of 100/100 and 1000/100 M, respectively. In contrast, the growth of *E. coli* is not affected by RA/SC. Therefore, RA/SC is a potentially bacteriostatic drug active to Vibrio species while doing low damage to natural bacteria in the gastrointestinal tract.

**Table 2 molecules-28-04243-t002:** Combined application of rosmarinic acid and derivatives with other antimicrobial agents.

Rosmarinic Acid with	Microorganisms	Synergy	References
Vancomycin	*Staphylococcus aureus*	+	[182]
Ofloxacin	*Staphylococcus aureus*	+	
Amoxicillin	*Staphylococcus aureus*	+	
Vancomycin	MRSA	+	
Ofloxacin	MRSA	−	
Amoxicillin	MRSA	−	
Penicillin	MRSA	+	[220]
Methyl rosmarinate	*Staphylococcus epidermidis* 5001	−	[77]
	*Stenotrophomonas maltophilia*	−	
	*Enterococcus faecalis C159-6*	−	
	*Staphylococcus lugdunensis T26A3*	−	
	*Pseudomonas aeruginosa ATCC 27583*	+	
	*Corynebacterium* T25-17	−	
	*Mycobacterium smegmatis 5003*	−	
	*Staphylococcus warneri* T12A12	−	
Isoquercetin	*Staphylococcus epidermidis* 5001	−	
	*Stenotrophomonas maltophilia*	−	
	*Enterococcus faecalis C159-6*	−	
	*Staphylococcus lugdunensis T26A3*	−	
	*Pseudomonas aeruginosa ATCC 27583*	−	
	*Corynebacterium* T25-17	−	
	*Mycobacterium smegmatis* 5003	−	
	*Staphylococcus warneri* T12A12	−	
Hyperoside	*Staphylococcus epidermidis* 5001	−	
	*Stenotrophomonas maltophilia*	−	
	*Enterococcus faecalis C159-6*	−	
	*Staphylococcus lugdunensis T26A3*	−	
	*Pseudomonas aeruginosa ATCC 27583*	−	
	*Corynebacterium* T25-17	−	
	*Mycobacterium smegmatis* 5003	+	
	*Staphylococcus warneri* T12A12	+	
Ulvan	*COVID-19*	+	[221]
Chitosan	*Escherichia coli*	+	[223]
Polyvlactic acid/layered double hydroxides-Rosmarinic acid	*Escherichia coli*	+	[224]
	*Staphylococcus aureus*	+	
Fe_III_/MoO_42_/PO_43_	herpes simplex virus	+	[225]
	VSV-Ebola pseudotypes	+	

An isoblot analysis revealed an antioxidant synergistic activity of rosemary extract in methanol and BHA, and it is found that rosemary extract in methanol and BHA interaction synergistically inhibit the growth of *S. aureus* and *E. coli*. As a result, rosemary extract increases the antioxidant effectiveness of BHA and BHT as well as the antibacterial impact of BHA, allowing for a 4.4- to 17-fold reduction in the amount of synthetic chemicals used [226].

There is a chance that bacterial dietary pathogens will induce intestinal diseases, but Madureira et al. [227] proved that, with a zeta potential of 20 to 30 mV, RA-loaded nanoparticles can stick to the intestinal epithelium and release the antimicrobial agent into the gut (against *L. innocua, B. cereus, E. coli* O157, *S. aureus*, *Y. enterocolitica*, and *S. typhimurium*).

In addition, a combination of rosmarinic acid, chicoric acid, and caffeic acid with metal (Fe_III_) as well as inorganic (MoO_42_ and Po_43_) ions was shown to be antiviral towards VSV-Ebola, herpes simplex virus, vaccinia viruses, and pseudotypes [225]. These combinations have antiviral activity, and their mode of action occurs at a very preliminary level of viral replication with limited cellular toxicity.

The antibacterial efficacy on two pathogens, *E. coli* (Gram-negative) and *Staphylococcus* (Gram-positive), was tested on polylactic acid/double-laminated hydroxides/rosmarinic acid (PLA/LDH-RA) films. Cicogna et al. [224] adopted the recommended ISO 22196:2011 procedure for this test, and a percentage of the index (antibacterial activity R), which allows the evaluation of the effectiveness of an antibacterial agent or therapy, is obtained by comparing the amount of bacteria present in the tested sample immediately after their inoculation and after a certain period of time, and the strains of *E. coli* and *S. aureus* had an antibacterial activity R of 2.56 log CFU/cm^2^ and 2.74 log CFU/cm^2^, respectively.

## 7. Cytotoxic Effect of RA

The demand for new antimicrobial products is high, and products derived from natural sources are seen as a promising solution including plant polyphenols, such as rosmarinic acid, which is a natural substance with antimicrobial activity [226,227].

The cytotoxic effect of free RA on the viability of HeLa and MCF-7 cells was evaluated by Fuster et al. [228], and the results show that the cytotoxicity of free RA was much weaker against both cell lines, These two cell lines are of human origin and have been widely used in cytotoxicity studies.

Kolettas et al. [229] have reported that RA failed to suppress hydrogen peroxide-induced apoptosis and did not possess antioxidant properties on Jurkat cells. RA has been reported to induce apoptosis and cause cytotoxicity in HepG2 cells [230,231].

On the other hand, Huang et al. [232] found that RA combined with adriamycin induced apoptosis in HepG2 cells, and the cytotoxic concentration of RA (1000 mM) increased MG132-induced apoptosis.

For example, Murakami et al. [233] indicated that the cytotoxicity of RA can be related to its pro-oxidant action, while Hur et al. [234] reported that RA increases reactive oxygen species and induces apoptosis in Jurkat cells and peripheral T cells via the mitochondrial pathway. In accordance with these publications, there are increases in protein oxidation, apoptosis, and cytotoxicity in MG132-treated HepG2 cells when the cytotoxic concentration of RA was applied [235].

## 8. Conclusions

Rosmarinic acid has been shown to be a strong antibacterial agent that may be able to stop the growth of a wide range of bacterial and fungal infections. It has been shown that it can stop a wide range of microbial species from making biofilm. Some ways that antifungals might work are by breaking down membranes and changing how mitochondria work. Because of its wide antibacterial range as well as its existence in naturally occurring bioactive compounds, rosmarinic acid needs to be investigated further in the quest to discover novel antibiotics. In addition, the increasing number of pharmacological research projects reveals the strong interest in the biological activities of RA, which are extremely diverse. Rosmarinic acid can be considered a rich source of potential candidates to be included in the food system with promising effects at predetermined concentrations, avoiding toxicity.

## Data Availability

Not applicable.

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
