# Peer review of "Application of Rosmarinic Acid with Its Derivatives in the Treatment of Microbial Pathogens"

_molecules, 2023, doi:10.3390/molecules28104243_

Round 1

Reviewer 1 Report

1.       Gram must be as capital letter G

2.       I suggest use IZ as abbreviation of inhibition zone in table.

3.       Improve the sharpness of figure 2

4.       In table 2 sometimes include , at least

5.       Use the same style, all genera or only abbreviation, preferred complete name of microbes

6.       Improve the conclusions include the perspectives.

Minor editing of English language required

Author Response

Dear reviewer

The authors would like to thank the editor of the Journal of Molecules for allowing us to revise the paper. We heartily appreciate the time and efforts dedicated by the reviewers for providing constructive feedback for our work which improved the paper quality significantly. We have addressed all these comments carefully and have made thorough corrections that we hope would meet the expectation of the journal. Those corrections are in red in the revised manuscript

Response to reviewer comments

  • Gram must be as capital letter G

Response: This point is taken into account and the word Gram is written with the capital letter G.

  • I suggest use IZ as abbreviation of inhibition zone in table.

Response: Done

  1. Improve the sharpness of figure 2

Response: Figure 2 sharpness is improved

4.In table 2 sometimes include , at least

Response: Done

5.Use the same style, all genera or only abbreviation, preferred complete name of microbes

Response: The full name of the microbes are included in the manuscript

  1. Improve the conclusions include the perspectives

Response: The conclusion is improved and the perspectives are included in the conclusion

  1. Minor editing of English language required

Response: English improved by native speakers

Reviewer 2 Report

Overall a comprehensive review RA antimicrobial activity.

There seems to be some inconsistence in the use of abbreviations for rosmarinic acid, sometime it is RA and other times it is AR.

 This review reports on the anti-microbial effects of rosma- 21 rinic acid and its associated derivatives, both in their free form and in combination with other microbial pathogens, and mechanisms of action. The last sentence of the introduction makes no sense when combination with other pathogens is referenced.

Line 143, how does resistance to rosemary make it an excellent antibacterial agent.

Line 147-148 makes no sense when antifungals mechanisms are referenced without any reference to RA. 

English and sentence structure ok, there are some areas where the sentence structure could be clearer. There are sections of the abstract and conclusion that make no sense,

Author Response

Dear reviewer

The authors would like to thank the editor of the Journal of Molecules for allowing us to revise the paper. We heartily appreciate the time and efforts dedicated by the reviewers for providing constructive feedback for our work which improved the paper quality significantly. We have addressed all these comments carefully and have made thorough corrections that we hope would meet the expectation of the journal. Those corrections are in red in the revised manuscript

Response to reviewer comments

  • There seems to be some inconsistence in the use of abbreviations for rosmarinic acid, sometime it is RA and other times it is AR

Response : Use of abbreviations for rosmarinic acid, is homogenized with RA

  • This review reports on the anti-microbial effects of rosma- 21 rinic acid and its associated derivatives, both in their free form and in combination with other microbial pathogens, and mechanisms of action. The last sentence of the introduction makes no sense when combination with other pathogens is referenced.

Response :The last sentence of the introduction is modified and improved

  • Line 143, how does resistance to rosemary make it an excellent antibacterial agent.

Response : Both Gram-positive and Gram-negative bacteria were found to be sensitive to rosemary,  and a mistake, the word "resistant" is changed to "sensitive".

  • Line 147-148 makes no sense when antifungals mechanisms are referenced without any reference to RA

Response : The sentence is deleted

5- Minor editing of English language required

Response: English improved by native speakers

Reviewer 3 Report

thank you for submitting your manuscript, It does look interesting to me but I would like you to do some basic revisions before I can send this out for peer review.

Firstly there are many problems with the English language, please can improve this either by getting a native speaker or similar to proof read and review the English language

1- start the introduction about rosmarinic acid and the problem of the work

2- It is good to add information on the traditional usage of this compound and its source in the introduction section 

3- giving information on the cytotoxicity of this compound 

4-  Conclusion  very short 

5- The authors should improve the introduction, making clearer what is the gap in the literature that is filled with this study. 

6- Methods: This section is completely missing: the study is not a systematic review because it also included review papers. It is not a narrative review because it included quantitative analysis. It is not an umbrella review because it also included original studies. It is not a scoping review because it tested a specific research question and pooled quantitative data. What kind of review is it? How was the paper selected? What were the criteria of inclusion? All these issues must be addressed and specified.

7- The Authors should add more practical recommendations for the reader, based on their findings.

need some corrections.

Author Response

Dear reviewer

The authors would like to thank the editor of the Journal of Molecules for allowing us to revise the paper. We heartily appreciate the time and efforts dedicated by the reviewers for providing constructive feedback for our work which improved the paper quality significantly. We have addressed all these comments carefully and have made thorough corrections that we hope would meet the expectation of the journal. Those corrections are in red in the revised manuscript

Response to reviewer comments

  1. Firstly there are many problems with the English language, please can improve this either by getting a native speaker or similar to proof read and review the English language

Response : The English language of the manuscript is proofread by a native speaker.

  1. start the introduction about rosmarinic acid and the problem of the work

Response : The introduction has been improved

  1. It is good to add information on the traditional usage of this compound and its source in the introduction section 

Response :  Information about the use of rosmarinic acid and its source is described in the following sections of Rosmarinic acid derivatives and Antimicrobial activity.

  1. giving information on the cytotoxicity of this compound 

Response :   Information on the cytotoxicity of rosmarinic acid is added to the manuscript in the section "Cytotoxic Effect of RA."

  1. Conclusion  very short 

Response : The conclusion  has been improved

  1. Minor editing of English language required

Response: English improved by native speakers

Round 2

Reviewer 3 Report

some notes must be performed 

1- add reference to you new addition:

The demand for new antimicrobial products is high and products derived from natural  sources are seen as a promising solution [Al-Rajhi et al. 2023; Qanash et al. 2023] including plant polyphenols, such as rosmarinic  acid, which is a natural substance with antimicrobial activity

Al-Rajhi, A.M.H.; Qanash, H.; Bazaid, A.S.; Binsaleh, N.K.; Abdelghany, T.M. Pharmacological Evaluation of Acacia nilotica Flower Extract against Helicobacter pylori and Human Hepatocellular Carcinoma In Vitro and In Silico. J. Funct. Biomater. 202314, 237. https://doi.org/10.3390/jfb14040237

Qanash, H.; Bazaid, A.S.; Aldarhami, A.; Alharbi, B.; Almashjary, M.N.; Hazzazi, M.S.; Felemban, H.R.; Abdelghany, T.M. Phytochemical Characterization and Efficacy of Artemisia judaica Extract Loaded Chitosan Nanoparticles as Inhibitors of Cancer Proliferation and Microbial Growth. Polymers 202315, 391.

2- All species please  write the full name then 1st liter only example Candida albicans, Escherichia coli and Staphylococcus aureus change to C. albicans etc 

3- Typhimurium  please change to typhimurium  etc 

4- what these : Moreno, Scheyer, Romano and Vojnov [181] please write only family name of reference  etc 

5- SaSalmonella had MIC and MBC of 0.9 and 1.0 mg/ml, respectively. please determine the type of tested compound in these sentence mg of ......

6- add a reference in : Recent research has revealed that natural substances based on secondary metabolites from plants can prevent the development of biofilms, which are responsible for about 80% of bacterial diseases.

some notes must be performed 

1- add reference to you new addition:

The demand for new antimicrobial products is high and products derived from natural  sources are seen as a promising solution [Al-Rajhi et al. 2023; Qanash et al. 2023] including plant polyphenols, such as rosmarinic  acid, which is a natural substance with antimicrobial activity

Al-Rajhi, A.M.H.; Qanash, H.; Bazaid, A.S.; Binsaleh, N.K.; Abdelghany, T.M. Pharmacological Evaluation of Acacia nilotica Flower Extract against Helicobacter pylori and Human Hepatocellular Carcinoma In Vitro and In Silico. J. Funct. Biomater. 202314, 237. https://doi.org/10.3390/jfb14040237

Qanash, H.; Bazaid, A.S.; Aldarhami, A.; Alharbi, B.; Almashjary, M.N.; Hazzazi, M.S.; Felemban, H.R.; Abdelghany, T.M. Phytochemical Characterization and Efficacy of Artemisia judaica Extract Loaded Chitosan Nanoparticles as Inhibitors of Cancer Proliferation and Microbial Growth. Polymers 202315, 391.

2- All species please  write the full name then 1st liter only example Candida albicans, Escherichia coli and Staphylococcus aureus change to C. albicans etc 

3- Typhimurium  please change to typhimurium  etc 

4- what these : Moreno, Scheyer, Romano and Vojnov [181] please write only family name of reference  etc 

5- SaSalmonella had MIC and MBC of 0.9 and 1.0 mg/ml, respectively. please determine the type of tested compound in these sentence mg of ......

6- add a reference in : Recent research has revealed that natural substances based on secondary metabolites from plants can prevent the development of biofilms, which are responsible for about 80% of bacterial diseases.

Author Response

Dear Reviewer

We would like to thank you for all the comments that have improved the quality of our work. We greatly appreciate the time and effort taken to review this work as well as the constructive comments that were provided which significantly improved the quality of the article. We have carefully considered all of these comments and have made extensive corrections that we hope will meet the journal's expectations. These corrections are indicated in red in the revised manuscript.

 Responses;

  1. add reference to you new addition:”The demand for new antimicrobial products is high and products derived from natural  sources are seen as a promising solution [Al-Rajhi et al.2023; Qanash et al. 2023] including plant polyphenols, such as rosmarinic  acid, which is a natural substance with antimicrobial activity”

Response : The proposed paragraph is taken and added to the text

  1. All species please  write the full name then 1st liter only example Candida albicans, Escherichia coli and Staphylococcus aureus change to  albicans etc

Response : The note is taken care of and all species names are written according to the requested suggestion

  1. Typhimurium  please change to typhimurium

Response: Done

  1. what these : Moreno, Scheyer, Romano and Vojnov [181] please write only family name of reference  etc 

Response :  the the reference "Moreno, Scheyer, Romano and Vojnov [181]" and other references are changed in this form "Moreno et al. [181]”

  1. SaSalmonella had MIC and MBC of 0.9 and 1.0 mg/ml, respectively. please determine the type of tested compound in these sentence mg of ......

Response :  Done

  1. add a reference in : Recent research has revealed that natural substances based on secondary metabolites from plants can prevent the development of biofilms, which are responsible for about 80% of bacterial diseases.

Response :  Done
